# The Relationship between Subjective Symptoms and Quality of Life in Conjunctivochalasis Patients

**DOI:** 10.3390/diagnostics11020179

**Published:** 2021-01-27

**Authors:** Aoi Komuro, Norihiko Yokoi, Hiroaki Kato, Yukiko Sonomura, Chie Sotozono, Shigeru Kinoshita

**Affiliations:** 1Department of Ophthalmology, Kyoto Prefectural University of Medicine, Kyoto 602-8566, Japan; akomuro@koto.kpu-m.ac.jp (A.K.); hiro-kat@koto.kpu-m.ac.jp (H.K.); yukky@ymail.plala.or.jp (Y.S.); csotozon@koto.kpu-m.ac.jp (C.S.); 2Department of Frontier Medical Science and Technology for Ophthalmology, Kyoto Prefectural University of Medicine, Kyoto 602-8566, Japan; shigeruk@koto.kpu-m.ac.jp

**Keywords:** conjunctivochalasis, dry eye, subjective symptoms, quality of life, questionnaire

## Abstract

The purpose of this study was to evaluate the differences in subjective symptoms between patients with conjunctivochalasis (CCh) and dry eye (DE), and examine the relationship between subjective symptoms and quality of life (QOL). In 75 eyes of 75 CCh patients and 122 eyes of 122 DE patients, 12 subjective symptoms classified into four groups depending on the mechanisms associated with symptoms (ITF: instability of tear film, IF: increased friction, R: reflex, and DTC: delayed tear clearance) were evaluated by use of a visual analogue scale (VAS). Fifteen items related to DE symptoms and their influence on daily life were evaluated by use of the dry eye-related quality-of-life score (DEQS) questionnaire, with overall degree of QOL impairment calculated as a QOL score. The correlation between the Total VAS score and QOL score were evaluated. Between the CCh and DE patients, significant differences in subjective symptoms were found in eye dryness, pain, tearing sensitivity to light, and heavy eyelids, while tearing was higher in CCh. A significant strong correlation was found between QOL score and Total VAS score, ITF, and IF in CCh patients. The QOL of CCh patients is strongly determined by decreased tear-film stability and increased friction during blinking.

## 1. Introduction

Typically, conjunctivochalasis (CCh) is a condition characterized by loose, redundant, binocular nonedematous conjunctival folds that occupy the lower tear meniscus [1,2]. Although CCh is sometimes observed in younger-age subjects, 98% or more of CCh cases are over 60 years old and the risk of acquiring CCh increases with age [3]. Many cases are asymptomatic. On the other hand, the subjective symptoms of CCh vary, since CCh causes precorneal tear film instability, increases the mechanical friction during blinking, and decreases tear flow at the lower tear meniscus [2,4,5,6]. Since CCh and dry eye (DE) commonly coexist, their subjective symptoms are similar. There are a few previous reports comparing the subjective symptoms between CCh and DE without CCh [5,7]. Pascuale et al. [7] reported that the subjective symptoms of a CCh patient were worsened by downgaze during reading or computer use and by vigorous blinking, while those of an aqueous deficient DE patient were worsened by upgaze due to increased interpalpebral exposure zone. Le et al. [5] reported that CCh was associated with an adverse impact on vision-related quality of life. There are four major mechanisms of ocular surface abnormalities, such as tear film instability, increased friction, reflex, and delayed tear clearance, and they can cause various DE-related subjective symptoms. Those mechanisms may also be involved in the subjective symptoms of CCh, since they are similar to those of DE. However, the exact details are currently unknown.

The DE-related quality-of-life score (DEQS) questionnaire is a validated diagnostic tool for DE in Japan [8]. It is comprised of 15 questions regarding bothersome ocular symptoms and their impact on daily life, including a question on mental health and stability within the previous week, and the overall degree of quality of life (QOL) impairment is calculated as a QOL score. This questionnaire has been administered in routine clinical practice and clinical studies in Japan since 2013 [9,10,11,12].

The purpose of this present study was to evaluate the differences in subjective symptoms between CCh and DE-disease patients using a visual analog scale (VAS). Moreover, in order to examine the relationship between subjective symptoms and their impact on QOL, the relationship between the VAS score and the QOL score of the DEQS questionnaire was examined.

## 2. Materials and Methods

The protocols of this study were approved by the Ethics Committee and the Institutional Review Board of Kyoto Prefectural University of Medicine, Kyoto (Approval No. RBMR-C-625-1, Approval Date: 25 November 2009), Japan, and were carried out in accordance with the tenets set forth in the Declaration of Helsinki. Prior written informed consent was obtained from all subjects after receiving a detailed explanation of the nature of the study and possible consequences associated with participation.

### 2.1. Subjects

This study involved 75 eyes of 75 CCh patients [16 males and 59 females; mean age: 67.8 ± 8.5 (mean ± SD) years]. CCh was diagnosed and graded according to the following classification. The lower lid margins were divided into three parts (nasal, middle, and temporal), and the severity of CCh was then graded (0: none; 1: mild; 2: moderate; and 3: severe) (Figure 1). Patients with a score of ≥1 at the middle area or ≥2 at the nasal or temporal areas were enrolled. This study also involved 122 eyes of 122 DE patients [14 males and 108 females; mean age: 66.2 ± 9.8 (mean ± SD) years] without CCh. The diagnostic criteria for DE were as follows: (1) DE symptoms and (2) a fluorescein breakup time (FBUT) of 5 s or less. Those who met two criteria (Japanese diagnostic criteria) [13] were enrolled as patients with DE. In all patients, the data of the eye with more severe subjective symptoms was used, yet if both eyes exhibited the same severity, the right eye data was used. Exclusion criteria were the use of antiglaucoma eye drops that may cause instability of tear film (ITF) and/or ocular surface epithelial damage, current contact lens wearers, and pregnancy, as well as subjects with recent changes to systemic medications affecting DE or vision and subjects who had received eye or lid surgery within the previous 6 months. In addition, all cases that the authors deemed inappropriate for being enrolled in the study were also excluded. All eyes with a meibomian gland dysfunction (MGD) diagnosis based on the Japanese diagnostic criteria for MGD [14] were strictly excluded from the analysis. Moreover, cases with DE accompanied by filamentary keratitis and/or superior limbic keratoconjunctivitis, or cases with symptoms that could be explained solely based on those abnormalities, were also excluded from the analysis under a combined agreement of the evaluators (A.K., N.Y., H.K., and Y.S.), as the purpose of this study was to specifically elucidate the relationship between CCh and DE in regard to subjective symptoms. All subjects were prohibited from using eye drops for at least 1 h prior to the examination in order to avoid any effect resulting from the instillation of the eye drops.

### 2.2. Severity of Subjective Symptoms Assessed by Visual Analogue Scale

Subjective symptoms at the time when a patient visits, including DE sensation, difficulty in opening the eye, foreign body sensation, pain, redness, tearing, discharge, itchiness, blurred vision, sensitivity to light, heavy eyelids, and eye fatigue, were evaluated by use of a visual analogue scale (VAS) (0: no symptoms; 100 mm: maximum symptoms) [15,16]. Those subjective symptoms were then divided into four groups depending on the mechanisms associated with subjective symptoms (ITF: instability of tear film (eye dryness, blurred vision, sensitivity to light, eye fatigue, heavy eyelids), IF: increased friction (difficulty in opening the eye, foreign body sensation, pain), R: reflex (redness, tearing), and DTC: delayed tear clearance (discharge, itchiness)) [17]. The Total VAS score and the sum of the VAS scores for each mechanism group were then calculated.

### 2.3. Severity of Subjective Symptoms Assessed by the Dry-Eye-Related Quality of Life Score (DEQS) Questionnaire

Fifteen items related to DE symptoms (irritation, eye dryness, pain, eye fatigue, heavy eyelids, redness) and influence on daily life (difficulty in opening the eye, blurred vision, sensitivity to light, eye-related problems when reading, eye-related problems when watching television or looking at a computer or cell phone, a feeling of distraction due to eye symptoms, eye symptoms affecting work, not feeling like going outside due to eye symptoms, depressed feeling due to eye symptoms) were evaluated by use of the DEQS questionnaire. Patients answered frequency (0 to 4, where 0 indicates that the respondent does not have the symptom and 4 indicates the highest frequency) and degree (1 to 4, with a larger number indicating a greater burden) of symptoms and disability during the week prior to the patient’s visit. The overall degree of QOL impairment is calculated as the QOL score (score: 0–100). The QOL score was calculated with the following formula: summary score = ((sum of the degree scores for all questions answered) × 25)/(total number of questions answered) [8].

### 2.4. Ocular Surface Examinations

For measurement of the FBUT and evaluation of corneal and bulbar conjunctival epithelial damage, a slit-lamp microscope with a cobalt blue filter and a blue-free filter was used [18]. After two drops of saline solution were instilled onto a fluorescein test strip (Showa Yakuhin Kako Co., Ltd., Tokyo, Japan), the strip was vigorously shaken and then softly touched to the central lower lid margin. After several natural blinks, the FBUT was counted as the time (in seconds) until the first appearance of a dark spot in the precorneal tear film when the eye was kept open. FBUT was measured three times, and then averaged. The van Bijsterveld scoring system [19] was used to score the cornea and conjunctival epithelial damage in each area from 0 to 3 depending on the severity of the staining, and the total score was then calculated. For the assessment of the corneal epithelial fluorescein staining, the cornea was divided into five areas. Staining was scored from 0 to 3 for each area, and the total score was then calculated (NEI score) [20]. The Schirmer 1 test (ST1, mm) without topical anesthesia was performed using a standard Schirmer test strip (AYUMI Pharmaceutical Co, Tokyo, Japan). The strip was placed for 5 min at the temporal one-third of the lower conjunctival fornix of the eye. The length (in mm) of the filter paper that had been wetted was then recorded.

### 2.5. Statistical Analysis

Statistical analyses were performed using JMP version 11.0 software (SAS Institute Inc., Cary, NC, USA). All results were expressed as mean ± standard deviation. Unpaired *t*-tests were used for statistical comparisons of the Schirmer 1 test value data, the FBUT, and the VAS scores. The Wilcoxon signed-rank test was used for ocular surface staining scores and CCh grade. The correlation between the Total VAS score, the VAS score of each clinical manifestation, and the QOL score were evaluated. Spearman’s rank correlation coefficients were used for the evaluation.

## 3. Results

### 3.1. Patient Background for CCh and DE

The patient background of CCh and DE is shown in Table 1. There were no significant differences in age, FBUT, the van Bijsterveld score, the NEI score, or the Schirmer 1 test value between the CCh and DE groups.

### 3.2. Subjective Symptoms

Subjective symptoms were stronger in the order of eye fatigue (56.1 ± 27.3), eye dryness (44.1 ± 30.3), and foreign body sensation (41.4 ± 27.1) in the CCh patients, and eye fatigue (60.7 ± 30.1), eye dryness (60.6 ± 29.1), and sensitivity to light (49.5 ± 35.1) in the DE patients. Between the CCh and DE patients, significant differences in subjective symptoms were found in eye dryness (44.1 ± 30.3 and 60.6 ± 29.1, respectively; *p* = 0.0002), pain (32.0 ± 29.9 and 42.7 ± 34.5, respectively; *p* = 0.023), tearing (28.6 ± 30.1 and 19.8 ± 25.9, respectively; *p* = 0.0065), sensitivity to light (39.9 ± 30.9 and 49.5 ± 35, respectively; *p* = 0.047), and heavy eyelids (34.9 ± 27.7 and 44.8 ± 33.2, respectively; *p* = 0.026) (Figure 2).

### 3.3. Relationship between Total VAS Score and QOL Score

A significant correlation was found between the Total VAS score and the QOL score in both CCh and DE (r = 0.75, r = 0.76, respectively; all: *p* < 0.0001). A significant strong correlation was found between the QOL score and ITF, IF (r = 0.79 and 0.70, respectively; all: *p* < 0.0001), a significant weak correlation was found between the QOL score and R (r = 0.27, *p* = 0.018), yet no significant correlation was found between the QOL score and DTC in CCh (Figure 3). A significant correlation was found between the QOL score and all ITF, IF, R, DTC in DE (r = 0.72, r = 0.71, r = 0.32, and r = 0.35, respectively; all *p* < 0.0001) (Figure 4) (Table 2).

## 4. Discussion

In this study, our findings revealed that the subjective symptoms of CCh were similar to those of DE. Tearing was the only symptom that was significantly higher than in DE. CCh disrupts the normal tear meniscus function, such as being a reservoir for retaining tears [21], the route for tears along the lid margin [22,23], and the delivery of tears to the ocular surface at the time of blinking [16]. Tearing is thought to occur from the blocking of tear flow at the tear meniscus by redundant conjunctiva [4,24,25]. Tearing also can occur episodically due to reflex tearing. That is the primary reason why eye dryness and tearing commonly coexist in CCh. Although most subjective symptoms can be improved via the standard topical treatment for DE [26], tearing may not respond to topical treatment. In our previous report, lacrimation improved in 36 of 37 eyes after surgical treatment for CCh [4]. Thus, it is important not to overlook the involvement of CCh in patients with tearing when planning treatment.

Subjects in this study were under treatment by the usual eye drop therapy with mixed preparations including preservative-free artificial tears 6 times/day, and/or 0.1% hyaluronic acid 4–6 times/day, and/or 3% diquafosol sodium (DQS) eye drops 4–6 times/day, and/or 2% rebamipide (RBM) ophthalmic suspension 4 times/day, and/or 0.1% fluorometholone 2 times/day depending on the observed fluorescein breakup pattern [26] and severity of symptoms. Eye dryness, pain, sensitivity to light, and heavy eyelids were significantly greater in DE than in CCh. Those symptoms are related to ITF, and this is the reason why eye drops typically used for treating DE, such as preservative-free artificial tears, hyaluronic acid eye drops, diquafosol sodium (DQS) eye drops, and/or rebamipide (RBM) ophthalmic suspension, were used to treat the CCh subjects in this study. In addition, fluorometholone was also used via the consideration of possible association with inflammation and the severity of symptoms. DQS eye drops reportedly increase aqueous fluid and mucins [27,28,29] which contribute to the increased tear film stability, while RBM reportedly can increase mucins and goblet cells [30,31] and is effective not only for the ITF mechanism in DE but also for friction-related disease (FRD) [32,33,34]. Therefore, the most appropriate selection of eye drops can be advised for CCh based on the primary mechanism and/or the predominant symptoms. For example, if ITF-related symptoms are primary, those symptoms should be treated with reference to BUPs, and if IF-related symptoms are primary, those symptoms should be treated with RBM. As was previously discussed, tearing is the secondary symptom due to ITF, and therefore, if tearing symptoms cannot be treated effectively via the use of eye drops for treating ITF, surgical treatment for CCh is the better indication.

The FBUT of the CCh group was as short as that in the DE group (i.e., both 2.1 s). One of the reasons for that is the fact that the ectopic meniscus adjacent to the redundant conjunctiva is responsible to the tear-film thinning [21] that leads to the tear-film breakup. Moreover, we speculate that undulating redundant conjunctiva forms numerous ectopic menisci, and the difference in the height of the meniscus is likely to cause uneven distribution of tears to the cornea, thus resulting in a breakup of the tear film. Another speculation is that the friction between the corneal surface and the eyelid margin during blinking may have a great impact on the surface structure of the corneal epithelial structure, such as a decrease of membrane-associated mucins that results in unstable tear film via the decreased wettability of the corneal surface. Moreover, Vu et al. [35] reported that when FRD, such as CCh, superficial limbic keratoconjunctivitis, or lid wiper epitheliopathy is present, the FBUT becomes significantly shorter as compared to DE with an absence of FRD.

In this present study, the DEQS score in the patients with CCh was 47.5, which was higher than previously reported. In a study by Sakane et al. [8], the authors reported a DEQS score of DE patients that was 33.7. Yokoi [17] previously described that tear-film instability and increased friction are the two important mechanisms for DE. Since CCh has both of those mechanisms, DE becomes apparent in eyes with CCh or when the existing DE becomes more severe. Hosotani [36] reported a high DEQS score (63.7) in benign essential blepharospasm (BEB) patients, probably due to the increased friction in BEB. Thus, there is a possibility that the subjective symptoms may worsen due to increased friction with CCh.

It should be noted that one limitation of this current study was that we did not evaluate the tear clearance or inflammation, as there are no noninvasive and quick tests for evaluating either of those. It has been reported that delayed tear clearance and inflammation are closely related to CCh, and, therefore, it is possible that they are associated with the subjective symptoms in CCh [7,37,38].

## 5. Conclusions

In conclusion, the findings in this study showed a relationship between subjective symptoms and the QOL of CCh patients, and that the QOL of CCh patients is strongly determined by tear-film instability and increased friction during blinking. Hence, further study is needed to elucidate the changes of QOL following surgical treatment for CCh.

## Figures and Tables

**Figure 1 diagnostics-11-00179-f001:**
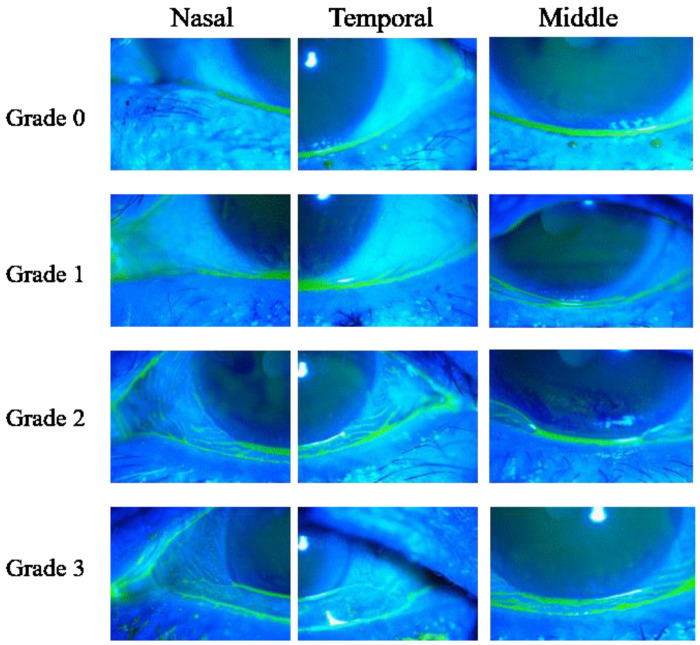
Images illustrating the grading system for conjunctivochalasis (CCh). Nasal and temporal area; Grade 0: no conjunctival fold, Grade 1: 2–3 conjunctival fords, Grade 2: several thick conjunctival folds, Grade 3: cystic or redundant conjunctival folds onto the lower lid margin. Middle area: Grade 0: no conjunctival fold, Grade 1: 2–3 folds appearing only after a forced blink, Grade 2: folds always existing with normal tear meniscus, Grade 3: cystic or redundant conjunctival folds onto the lower lid margin.

**Figure 2 diagnostics-11-00179-f002:**
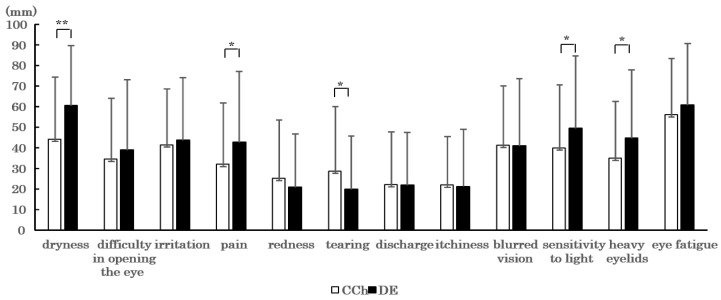
Mean visual analog scale (VAS) scores for the CCh and dry eye (DE) patients. Data were expressed as mean ± SD. * *p* < 0.05, ** *p* < 0.001 (paired *t*-test). A *p*-value of <0.05 was considered statistically significant.

**Figure 3 diagnostics-11-00179-f003:**
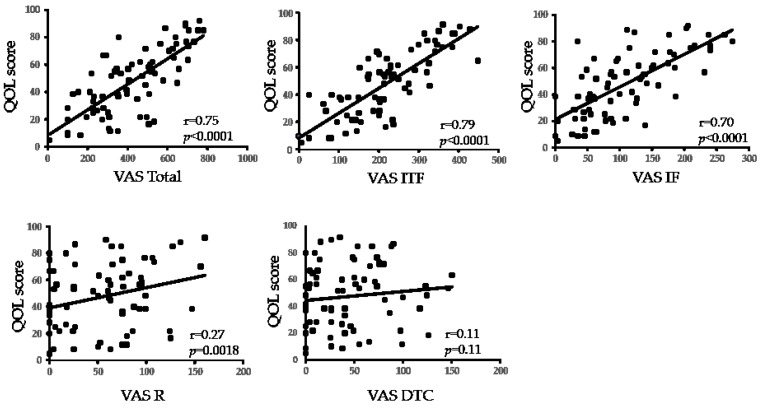
Correlation between VAS score and quality of life (QOL) score in CCh. “r” indicates the Spearman correlation coefficient. ITF, instability of tear film; IF, increased friction; R, reflex; DTC, delayed tear clearance.

**Figure 4 diagnostics-11-00179-f004:**
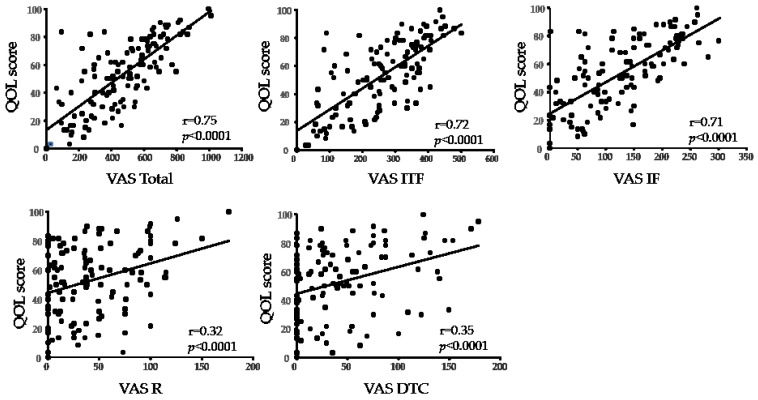
Correlation between VAS score and QOL score in DE. “r” indicates the Spearman correlation coefficient. ITF, instability of tear film; IF, increased friction; R, reflex; DTC, delayed tear clearance.

**Table 1 diagnostics-11-00179-t001:** Demographics and clinical characteristics of study subjects.

	CCh (*n* = 75)	DE (*n* = 122)
Age, years	67.8 ± 8.5	66.2 ± 9.9
Male:Female	16:59	14:108
FBUT, sec	2.1 ± 1.5	2.1 ± 1.4
van Bijsterveld score	1.3 ± 1.9	1.4 ± 1.5
NIH score	1.5 ± 2.7	1.8 ± 2.5
CCh grade	4.6 ± 1.7	1.5 ± 0.7 *
Schirmer 1, mm	11.2 ± 9.0	12.9 ± 10.6

* *p* < 0.0001.

**Table 2 diagnostics-11-00179-t002:** Correlation between QOL score and VAS score.

	Versus	r	*p*
QOL score of CCH	Total	0.75	*p* < 0.0001
	ITF	0.79	*p* < 0.0001
	IF	0.70	*p* < 0.0001
	R	0.27	*p* = 0.018
	DTC	0.11	*p* = 0.11
QOL score of DE	Total	0.75	*p* < 0.0001
	ITF	0.72	*p* < 0.0001
	IF	0.71	*p* < 0.0001
	R	0.32	*p* < 0.0001
	DTC	0.35	*p* < 0.0001

The combinations in which a significant correlation was observed are shown. “r” indicates the Spearman correlation coefficient. A *p*-value of <0.05 was considered statistically significant. ITF, instability of tear film; IF, increased friction; R, reflex; DTC, delayed tear clearance.

## Data Availability

The data presented in this study are available on request from the corresponding author.

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
