# Peer review of "The Relationship between Subjective Symptoms and Quality of Life in Conjunctivochalasis Patients"

_diagnostics, 2021, doi:10.3390/diagnostics11020179_

Round 1

Reviewer 1 Report

This study evaluated the differences in the subjective symptoms between patients with conjunctivochalasis (CCh) and dry eye (DE). The authors also examined the relationship between subjective symptoms and quality of life (QOL). The authors found that the significant differences in the subjective symptoms were found in eye dryness, pain, tearing sensitivity to light and heavy eyelids. The authors also found that a significant correlation between the total VAS score and the QOL score in both CCh and DE. The authors concluded that QOL of CCh patients is strongly determined by tear-film instability and increased friction during blinking.

  1. Based on the results of this study, please discuss the different treatment may apply on patients with different symptoms.

  1. Were these participants been treated during the study?

Author Response

Response to Reviewer 1 Comments

We wish to express our appreciation to the Reviewer for his/her insightful comments, which have helped us to significantly improve our manuscript.

This study evaluated the differences in the subjective symptoms between patients with conjunctivochalasis (CCh) and dry eye (DE). The authors also examined the relationship between subjective symptoms and quality of life (QOL). The authors found that the significant differences in the subjective symptoms were found in eye dryness, pain, tearing sensitivity to light and heavy eyelids. The authors also found that a significant correlation between the total VAS score and the QOL score in both CCh and DE. The authors concluded that QOL of CCh patients is strongly determined by tear-film instability and increased friction during blinking.

Point 1: Based on the results of this study, please discuss the different treatment may apply on patients with different symptoms.

Response 1: We greatly appreciate the Reviewer’s helpful comment. Please note that we have now added the following text to our revised manuscript (Page 7, Lines 204-219):

“Eye dryness, pain, sensitivity to light, and heavy eyelids were significantly greater in DE than in CCh. Those symptoms are related to ITF, and this is the reason why eye drops typically used for treating DE, such as preservative-free artificial tears, hyaluronic acid eye drops, diquafosol sodium (DQS) eye drops, and/or rebamipide (RBM) ophthalmic suspension, were used to treat the CCh subjects in this study. In addition, fluorometholone was also used via the consideration of possible association with inflammation and the severity of symptoms. DQS eye drops reportedly increase aqueous fluid and mucins [27-29] which contribute to the increased tear film stability, while RBM reportedly can increase mucins and goblet cells [30, 31] and is effective not only for the ITF mechanism in DE but also for friction-related disease (FRD) [32-34]. Therefore, the most appropriate selection of eye drops can be advised for CCh based on the primary mechanism and/or the predominant symptoms. For example, if ITF-related symptoms are primary, those symptoms should be treated with reference to BUPs, and if IF-related symptoms are primary, those symptoms should be treated with RBM. As was previously discussed, tearing is the secondary symptom due to ITF, and therefore, if tearing symptoms cannot be treated effectively via the use of eye drops for treating ITF, surgical treatment for CCh is the better indication.”

Point 2: Were these participants been treated during the study?

Response 2: We greatly appreciate the Reviewer’s helpful comment. Please note that we have now added the following statement regarding the treatment (Page 7, Lines 200-204).

Subjects in this study were under treatment by the usual eye drop therapy with mixed preparations including preservative-free artificial tears 6 times/day, and/or 0.1% hyaluronic acid 4-6 times/day, and/or 3% diquafosol sodium (DQS) eye drops 4-6 times/day, and /or 2% rebamipide (RBM) ophthalmic suspension 4 times/day, and/or 0.1% fluorometholone 2 times/day depending on the observed fluorescein breakup pattern [26] and severity of symptoms.”

Reviewer 2 Report

The inclusion and exclusion criteria are quite vague, and this is important to clarify (Rx used, both systemic and topical, co-morbidities. etc)

VAS used- at what timepoint were subject tested? Was it for the specific scores during their visit, or was it during last day/week/month?

Author Response

Response to Reviewer 2 Comments :

We wish to express our appreciation to the Reviewer for his/her insightful comments, which have helped us to significantly improve our manuscript.

Point 1: The inclusion and exclusion criteria are quite vague, and this is important to clarify (Rx used, both systemic and topical, co-morbidities. etc)

Response 1: We greatly appreciate the Reviewer’s helpful comment. Please note that we have now added the following statement in our revised manuscript regarding the exclusion criteria (Page 2, Lines 71-76).

“Exclusion criteria were the use of antiglaucoma eye drops that may cause ITF and/or ocular surface epithelial damage, current contact lens wearers, and pregnancy, as well as subjects with recent changes to systemic medications affecting DE or vision and subjects who received eye or lid surgery within the previous 6 months. In addition, all cases that the authors deemed inappropriate for being enrolled in the study were also excluded. ”

Point 2:
VAS used- at what timepoint were subject tested? Was it for the specific scores during their visit, or was it during last day/week/month?

Response 2: We greatly appreciate the Reviewer’s comment. Please note that we have now added the following statement regarding the time-point in section 2.2. Severity of Subjective Symptoms Assessed by Visual Analogue Scale (Page 3, Line 94):

"... at the time when a patient visits, ..."

Please note that we also added the following statement regarding the time-point in section 2.3. Severity of Subjective Symptoms Assessed by the Dry-Eye-Related Quality of Life Score (DEQS) Questionnaire (Page 3, Line 113 through Page 4, Line 114):

“... during the week prior to the patient's visit."